# Impacts of the Attachment and Child Health (ATTACH^TM^) Parenting Program on Mothers and Their Children at Risk of Maltreatment: Phase 2 Results

**DOI:** 10.3390/ijerph20043078

**Published:** 2023-02-09

**Authors:** Nicole Letourneau, Lubna Anis, Jason Novick, Carrie Pohl, Henry Ntanda, Martha Hart

**Affiliations:** 1Owerko Centre for Children’s Neurodevelopment and Mental Health, Alberta Children’s Hospital Research Institute, Faculty of Nursing, and Cumming School of Medicine, Departments of Pediatrics, Psychiatry and Community Health Sciences, University of Calgary, Calgary, AB T2N 1N4, Canada; 2Owerko Centre for Children’s Neurodevelopment and Mental Health, Alberta Children’s Hospital Research Institute, University of Calgary, Calgary, AB T2N 1N4, Canada

**Keywords:** ATTACH^TM^, parental reflective function, parenting intervention, child development

## Abstract

Early adversity (e.g., family violence, parental depression, low income) places children at risk for maltreatment and negatively impacts developmental outcomes. Optimal parental reflective function (RF), defined as the parent’s ability to think about and identify thoughts, feelings, and mental states in themselves and in their children, is linked to secure attachment and may protect against suboptimal outcomes. We present the results of Phase 2 randomized control trials (RCTs) and quasi-experimental studies (QES) of the Attachment and Child Health (ATTACH^TM^) parental RF intervention for families with children at risk for maltreatment. Phase 2 parents experiencing adversity, along with their children aged 0–5 years (*n* = 45), received the 10–12-week ATTACH^TM^ intervention. Building on completed Phase 1 pilot data, Phase 2 examined outcomes of long-standing interest, including parental RF and child development, as well as new outcomes, including parental perceived social support and executive function, and children’s behavior, sleep, and executive function. RCTs and QES revealed significant improvements in parents’ RF, perception of social support, and executive function, children’s development (i.e., communication, problem-solving, personal–social, and fine motor skills), and a decrease in children’s sleep and behavioral problems (i.e., anxiety/depression, attention problems, aggressive behavior, and externalizing problems), post-intervention. ATTACH™ positively impacts parental RF to prevent negative impacts on children at risk of maltreatment.

## 1. Introduction

Early adversity (e.g., family violence, parental depression, low income) places children at risk for maltreatment, such as abuse and neglect [1,2], and negatively impacts crucial social, cognitive, genetic, and neurologic developmental outcomes [3,4,5,6,7,8]. In contrast, secure attachment between parents (or primary caregivers) and children, underpinned by parental reflective function (RF), is linked to positive health and developmental outcomes in children [9,10,11,12]. Parental RF is defined as the parent’s ability to think about and identify thoughts, feelings, and mental states in themselves and in their children [13,14]. According to leading theorist Peter Fonagy, RF operationalizes mentalizing, which involves attending to mental states in oneself and others and interpreting behavior accordingly [13,15]. RF differs from related terms such as empathy, emotional intelligence, metacognition, mindblindness, mindfulness, mindreading, and theory of mind, [15,16,17]. While not synonymous with RF, they may tap the same underlying neurobiological socio-cognitive system as RF and focus on internal representations of the child [17,18,19]. Parental RF is RF that is specific to the parent–child relationship and most relevant to the risk of child maltreatment [20].

Impairments in parental RF are associated with insecure attachment and disorganization in infants and increased vulnerability to psychological disorders in childhood, adolescence, and adulthood [21,22,23,24,25]. Parents who are highly reflective interpret their child’s behavior by considering their own mental states and those of their children as well as interactions between each other’s mental states [10,11,13], while those with low RF tend to be unaware of their own internal experiences and/or their children’s mental states [13,19]. Parental RF has been linked to parent–child interaction quality, with higher RF associated with sensitive parenting and lower RF associated with disruptive parenting quality [11,25,26,27,28,29].

The ability of parents to be reflective during interactions with their children is often observed to be suboptimal in populations at risk for maltreatment, including those affected by parental histories of early trauma or adversity, current drug and alcohol abuse, psychopathology, and/or dysfunctional parenting [3,7,30,31,32]. High-risk families are also likely to be exposed to intersecting socio-economic risks, including low maternal education, single parenthood, young age at conception, and minority status [33]. Moderate and higher RF ability in mothers predicts secure infant attachment in both low [10,19,34] and high-risk [31,35] samples.

Parental RF appears modifiable [26,27,36,37,38,39,40,41]; thus, early intervention may foster secure attachment and healthy child development. A recent systemic review [36] of dyadic interventions targeting improvement in parental RF reported a significant reduction in disorganized attachment in infants (risk ratio: 0.50; 95% CI [0.27, 0.90]) [36]. Parental RF interventions also improved RF for mothers experiencing addiction [27,37,38,39,40], imprisonment [42,43,44], depression, family violence, and/or poverty [11,26,45]. Despite variations in intervention design, sample size, and coding methods for assessing parental RF, RF interventions appear to enhance maternal RF in high-risk families [46].

### 1.1. Parental RF and Executive Function of Parents and Children

A set of processes that serve to control behavior directed toward a goal constitutes executive function which includes attention, working memory, task-switching (or set shifting), and inhibitory control [47]. These mental abilities enable an individual to control their emotions and behavior to achieve their goals [48,49]. Compared to mothers with less optimal working memory, those with better working memory are more capable of managing their own emotions and behavioral responses to their distressed infants [50,51,52,53,54], display more interest in their child’s feelings and curiosity [55], and more positive behaviors toward their children in a frustration-based task [56,57]. Mothers with better set-shifting capacities show higher levels of interest and curiosity in their children’s thoughts and feelings and adjust their behaviors during dyadic interactions to accommodate their children, consistent with higher parental RF [58].

Emotional self-regulation has been identified as a key component of RF [51,58] and likelyequips parents to better respond to their children’s emotions [59]. Parents with high RF are more likely to show resilience in coping with emotional distress instead of reacting impulsively [13,60]. Self-regulation of extreme emotions may help parents adjust their behaviors to match their children’s needs [61,62] and support regulation of their children’s affective states [59]. Executive function can be improved through training or interventions, such as mindfulness meditation in adults [49] and children [50,52]. However, it is unclear to what extent these improvements are generalizable to parenting and how they translate into positive changes in RF or children’s behavior or executive function [50]. To our knowledge, no studies have examined the impact of interventions focused on parental RF on parents’ and children’s executive function.

### 1.2. Parental RF and Child Development

Parental RF is positively associated with the development of children’s cognitive and social skills [23,63] during infancy and early childhood [11,13,64]. Parental RF interventions have demonstrated impacts on personal social domains of child development but not other domains, such as communication, problem-solving, and gross and fine motor skills [11]. Other interventions focused on parents’ internal representations of the child (similar to RF or mentalization) in the context of the parent–child relationship (e.g., parent–infant psychotherapy) did not demonstrate impacts on either child development [65] or vocabulary [66,67]. These studies had limitations, such as small sample sizes [11,65] and lack of blind assessments [65]. Thus, further research is required to determine the effects of the intervention on multiple domains of children’s development.

### 1.3. Parental RF and Children’s Behavioural and Sleep Problems

Parents with a higher RF perceive their children’s behavior as an expression of their children’s mental states [44]. A significant association was found between mothers’ higher RF and children’s improved social-emotional skills, and fewer socio-emotional and behavioral problems [68,69,70,71,72,73]. In contrast, children of parents with low parental RF are more likely to display emotion dysregulation, anxiety disorders, attention-deficit hyperactivity disorder (ADHD) [74], internalizing and externalizing behaviors [75], and lower social-emotional competencies [73,76]. Minding the Baby, an RF-based, trauma-informed, preventive home-visiting intervention for high-risk mothers and their children, reduced externalizing behaviors (e.g., aggression, hyperactivity) in children [77]. Another RF-based parenting intervention for foster parents, Family Minds, reported no significant changes in children’s internalizing behaviors (e.g., anxiety, social withdrawal) post-intervention [78,79]. These mixed findings suggest a need for further study of the impacts of parental RF-based interventions on both internalizing and externalizing behavioral problems in children at risk for maltreatment.

Additionally, infants’ sleep problems are associated with both maltreatment and adverse developmental outcomes [80,81,82]. Compared to parents with higher RF levels, parents with lower RF levels tend to be less emotionally available and committed to their children, are less capable of tolerating or regulating their children’s signals of distress (e.g., that signal a need for sleep), and are prone to being absorbed by their own experiences [80]. Infants’ sleep problems evoke more negative reactions and behavior (e.g., anger) in parents with lower levels of parental RF [83]. Parent–infant psychotherapy focusing on mothers’ internal representations of their infants was more effective than usual care in reducing disordered infant sleep [82,83,84]. Little is known about the effectiveness of parental RF interventions on infant sleep in children at risk for maltreatment.

### 1.4. Parental RF and Parents’ Perceived Social Support

Perceived social support is defined as an individual’s sense of the availability of their friends, family members, and others to provide material, psychological, and emotional aid during times of need [85,86]. The more stress parents experience in their close relationships (e.g., having no support or low support), the more challenging it may be to focus on their children’s mental states and behaviors [85,87]. Parental RF may impact interrelatedness with others in the social network, which may increase the perception of the quality and quantity of social support [85]. Yet no studies have examined the effectiveness of parental RF-based interventions on maternal perception of social support.

### 1.5. ATTACH™ for Children at Risk of Maltreatment

Parental RF-based parenting interventions may be among the most effective strategies to help vulnerable families facing adversities that put infants/children at risk for maltreatment and associated poor outcomes [11,26,36,45,88]. One such intervention is the Attachment and Child Health (ATTACH^TM^) program, a psycho-educational intervention designed to improve children’s development, health, and relationship outcomes [11,26,45,89,90]. ATTACH^TM^ has been recognized as one of Harvard University’s Frontier of Innovation Projects [91]. Phase 1 findings from the evaluations of ATTACH™-employing randomized controlled trials (RCTs) and quasi-experimental studies (QES) revealed that ATTACH™ significantly improved: (a) parents’ RF [26,45]; (b) parent–child interaction quality [11]; and (c) children’s development, specifically in the personal social domain [11]. Data that combined samples across Phase 1 and Phase 2 revealed that intervention improved the likelihood that children were securely attached [45], and data from one pilot study that collected blood samples in Phase 2 showed that ATTACH™ improved mothers’ and children’s immune cell gene expression linked to inflammation [90].

### 1.6. Research Aims of Current Study

The current report aims to build on findings to date from Phase 1 and Phase 2 pilots, this time to determine the impact of ATTACH^TM^ on outcomes of long-standing interest, including parental RF and child development, as well as new outcomes examining parents’ executive function and perceived social support and children’s behavioral problems, sleep, and executive function. We hypothesize that families who receive the ATTACH^TM^ intervention will demonstrate improved parental RF and child development outcomes, consistent with past findings, as well as improved executive function and perceptions of social support in parents and reduced behavioral and sleep problems, and improved executive function in children.

## 2. Materials and Methods

Guided by the Innovate, Develop, Evaluate, Adapt, Scale (IDEAS; [3,7,92]) Impact Framework that emphasizes rapid cycling trial methods, Phase 1 (now completed) involved pilots #1 to #3. Phase 2 of the ATTACH^TM^ intervention testing involved four pilot studies (#4 to #7) employing RCTs and QES methods. Specifically, two RCTs (pilot studies #4 and #6) tested ATTACH™ with parents and children who received service from inner-city agencies in Calgary, Alberta, serving low-income families and families affected by domestic violence. The QES (pilot studies #5 and #7) were conducted to provide the ATTACH^TM^ intervention to control group participants from each RCT after each RCT was completed. The Conjoint Health Research Ethics Board at the University of Calgary, Alberta, Canada, approved the study to conduct Phase 2 of the ATTACH^TM^ intervention. All participants underwent a process of informed consent.

### 2.1. Outline of the ATTACH^TM^ Intervention

The ATTACH^TM^ intervention is a brief parenting psycho-educational intervention with dyadic (mother/primary caregiver and infant) and triadic (mother/primary caregiver, infant, and co-parenting support person) components that foster parental RF [26,45,89]. The intervention is designed to help parents enhance and develop their capacity for RF (or mentalizing). Accordingly, the ATTACH^TM^ intervention aids in developing the capacity to think about mental states and to consider how one’s own mental states might affect others and how others’ mental states might have an impact on oneself. The format consists of 10–12 weekly face-to-face dyadic intervention sessions with the parent (or primary caregiver) and an ATTACH^TM^ facilitator and 2–3 face-to-face triadic intervention sessions with the parent or primary caregiver, co-parent, and an ATTACH^TM^ facilitator, that last approximately 60 min. When working with the families, the facilitator focuses on building a therapeutic relationship as well as engaging in RF techniques. Each one-on-one dyadic session involves videotaped interaction of the participating parent playing with their child, followed by a feedback session. Parents are also asked to consider hypothetical and real-life mildly stressful situations that require RF skills. For example, the hypothetical situation from session #1 asks parents to consider family members’ thoughts and feelings during mealtime when the child drops their food on the floor. Real-life situations derive from parents’ stressful experiences over the past week, in which parents are asked to think about the thoughts and feelings of everyone involved. Triadic sessions are similar to dyadic sessions with the inclusion of the co-parent but do not include a real-life situation task. Once the therapeutic relationship is established (after 6 one-on-one sessions of therapy), the caregiver invites their co-parenting support person to take part in 2–3 triadic sessions (at sessions 7, 9, and 11).

### 2.2. Samples

Staff members of the participating inner-city agencies (a low-income family service agency and a women’s shelter) recruited a convenience sample of parents for ATTACH^TM^. Fourteen primary caregivers (13 mothers and 1 grandmother), each with one participating infant/young child under 36 months of age, participated in the first RCT (ATTACH^TM^ pilot study #4) at the low-income family service agency. Participants were randomly assigned to intervention (*n* = 7) or control group (*n* = 7) using an online randomizer and opaque sealed envelopes. Another sample of 20 parents (all mothers), each with one participating infant/young child under 6 years of age, participated in the second RCT (ATTACH^TM^ pilot study #6) at the women’s shelter. Participants were also randomly assigned to groups, as above (*n* = 10 intervention; *n* = 10 control). QES was conducted to provide the ATTACH^TM^ intervention to control group participants from each RCT. To elaborate, the waitlist control group from the first RCT (ATTACH^TM^ pilot study #4; *n* = 7) received the ATTACH^TM^ intervention later and comprised the QES (ATTACH^TM^ pilot study #5). Similarly, the waitlist control group from the second RCT (ATTACH^TM^ pilot study #6; *n* = 10) received the ATTACH^TM^ intervention later, comprising the QES (ATTACH^TM^ pilot study #7). Post-assessments from the RCTs (ATTACH^TM^ pilot studies #4 and #6) were used as the baseline assessments for the QES (ATTACH^TM^ pilot studies #5 and #7). All participants ultimately completed the ATTACH^TM^ intervention.

### 2.3. Measures

We collected demographic data from participants pertaining to ethnicity, first language, marital status, education, employment, and age of parents and children at baseline. Furthermore, data on the parental outcome measurements (parental RF, perceived social support, executive function) and child outcome measurements (development, behavioral problems, sleep, and executive function) were collected at baseline and post-intervention. We selected measures to align with the recommendations of the Harvard Center of the Developing Child’s Frontier of Innovation program [91].

#### 2.3.1. Parental Outcome Measures

Parental Reflective Functioning Questionnaire (PRFQ). The PRFQ [93] is an 18-item self-report questionnaire that examines the extent to which parents demonstrate RF in parenting. The PRFQ is designed to be administered to parents with infants/children 0–5 years of age. The PRFQ subscales examine parents’: (1) pre-mentalizing modes (lower scores are more positive); (2) certainty about mental states (higher scores are more positive); and (3) interest and curiosity about mental states (higher scores are more positive). Each item of the PRFQ is answered on a Likert scale rating from 1 (strongly disagree) to 7 (strongly agree). The PRFQ pre-mentalizing, certainty about mental states, and interest and curiosity subscales demonstrated excellent construct validity, internal consistency, and reliability [20].

Five Facet of Mindfulness Questionnaire (FFMQ). The FFMQ [94,95] is a 39-item self-administered questionnaire examining the prevalence of mindfulness, as a concept related to RF, among parents. Observing, describing, acting with awareness, non-judgment of inner experiences, and non-reactivity to those experiences are the five facets of mindfulness. Responses to each item range from 1 (never true) to 5 (often true). Higher scores are consistent with more mindfulness. Studies have demonstrated excellent construct validity [95], favorable discriminant validity and convergent validity [95,96], and excellent internal consistency as well as incremental validity [95] of the FFMQ.

Behavior Rating Inventory of Executive Function—Adult Version (BRIEF-A). The BRIEF-A [97] is a 75-item self-administered questionnaire that captures adults’ executive function or self-regulation in the everyday environment. It provides an overall measure in the Global Executive Composite (GEC) score, based on items assessing inhibition, self and task monitoring, planning, initiative, emotional control, working memory, and organization. Higher scores indicate greater impairment in executive function. Responses to each item range from 1 (never) to 3 (often). The BRIEF-A has demonstrated excellent internal consistency and convergent validity [98].

Social Support Effectiveness Questionnaire (SSE-Q). The SSE-Q [99] assesses parents’ appraisal of support from the person most relied upon during the three months prior to filling out the questionnaire. It is a 25-item scale that provides respondents with a brief description of three types of social support (instrumental, informational, and emotional) and asks them to assess the degree to which the: (1) quantity of support matched the amount desired, (2) they wished the support was offered differently, (3) they felt the support was offered skillfully, (4) they felt support was difficult to get, and (5) others offered support without being asked. Responses to each item ranged from 0 (never) to 4 (always). Higher scores are more positive. The SSE-Q is well validated, and exhibits demonstrated excellent internal consistency (α = 0.95) [100].

#### 2.3.2. Child Outcome Measures

Ages and Stages Questionnaire, third edition (ASQ-3). The ASQ-3 [101] assesses children’s global development [102] with reference to five domains: (1) communication, (2) personal–social, (3) problem-solving, (4) fine motor, and (5) gross motor. The questionnaire is a series of parent-completed, age-specific, questionnaires and is used to identify children with developmental delays. Different versions are available for children ranging in age from 1 month to 66 months. Higher scores are consistent with better performance. The ASQ-3 has demonstrated excellent validity (between 0.82–0.88), a sensitivity of 86%, and a specificity of 85% [103].

Child Behavior Checklist (CBCL). The CBCL for Children Ages 1.5–5 [104] is a 100-item parent-report questionnaire measuring perceptions of their children’s behavior within the preceding 2 months. The CBCL examines internalizing problems (i.e., emotionally reactive, anxious/depressed, somatic complaints, and withdrawn) and externalizing problems (i.e., aggressive behavior, attention problems). The extent of internalizing and externalizing problems is examined on a scale ranging from 0 (not true) to 2 (very true or often true). Items also assess aspects of sleep, such as whether one’s child sleeps less or more than others and has difficulty sleeping. Higher scores indicate a greater risk for behavioral or sleep problems. The CBCL has excellent convergent validity [104] and displays high internal consistency, with α = 0.87 and 0.89 for the internalizing and externalizing problem scales, respectively [105,106].

Behavior Rating Inventory of Executive Function—Preschool Version (BRIEF-P). The BRIEF-P [107] is a 63-item parent-report measure of the behavioral manifestations of children’s executive function within the context of the everyday home environment. Items measure inhibition, shifting of attention, emotional control, working memory, and planning. Responses to each item range from 1 (never) to 3 (often). The Global Executive Composite score ranges from 63 to 189, with higher scores indicating a higher propensity of problematic child behaviors, consistent with less effective executive function. The BRIEF-P has demonstrated excellent internal consistency [108], test-retest reliability [109], and content validity [110].

### 2.4. Data Analysis

Demographic data were analyzed with measures of central tendency and frequencies. Sample characteristics of the intervention and control groups in pilot studies #4 and #6 at baseline were examined with chi-square tests to determine group equivalency. Independent samples *t*-tests were employed to evaluate differences between the intervention group and control group at post-intervention for the RCTs (ATTACH^TM^ pilot studies #4 and #6). Paired samples *t*-tests were conducted to examine changes in outcome measures from baseline to post-intervention for the QES (ATTACH^TM^ pilot studies #5 and #7). For the purpose of reporting program outcomes and giving directional hypotheses, one-tailed testing at the 0.05 level was used to report statistically significant outcomes. Effect sizes (Cohen’s d) were also calculated with values of 0.20, 0.40, and 0.70, representative of small-, medium-, and large-effect sizes, respectively. Participants’ data were analyzed according to their original group assignment as per intention-to-treat principles.

## 3. Results

### 3.1. Sample Characteristics

#### 3.1.1. ATTACH^TM^ Pilot Studies #4 and #5

The 14 caregivers (mostly mothers) in ATTACH^TM^ pilot study #4 were from the low-income family service agency and were approximately 31 years of age at enrolment, on average. The majority spoke English as a first language, were single caregivers, born in Canada, and had a high school diploma as their highest educational attainment. The most common ethnicity was Caucasian, and the majority were working full-time. They indicated a moderate level of social support on the SSE-Q. Refer to Table 1 for the caregivers’ demographic characteristics from ATTACH^TM^ pilot study #4.

Eight male and six female children participated in pilot study #4 and were an average of 25 months of age at enrolment. English was the primary language spoken by all of the children, and the majority were Caucasian. According to the subscale cut-off scores of the CBCL, on average, children did not demonstrate problematic behaviors related to emotional reactivity, anxiety/depression, somatic complaints, withdrawal, attention, aggression, or sleep, at baseline. Children’s development was also on schedule at enrolment, with the average ASQ-3 subscale scores above the cut-offs for communication, gross motor skills, fine motor, problem-solving, and personal social skills. Please refer to Table 2 for the child demographic characteristics from ATTACH^TM^ pilot study #4. A total of seven caregivers participated in ATTACH^TM^ pilot study #5, which included the control group from ATTACH^TM^ pilot study #4.

#### 3.1.2. ATTACH^TM^ Pilot Studies #6 and #7

The 20 mothers who participated in ATTACH^TM^ pilot study #6 from the women’s shelter were approximately 32 years at enrolment, on average. The majority spoke English as a first language, were single, born in Canada, and had attained some college as their highest level of education. The most common ethnicity was Indigenous Canadian, and the majority of caregivers were unemployed/not in the workforce. Caregivers indicated a moderate level of social support on the SSE-Q. Please refer to Table 3 for the caregiver demographic characteristics from pilot study #6.

Nine male and 11 female children participated in pilot study #6 and were an average of 31 months of age at enrolment. The majority of children spoke English as a primary language. The most common ethnicities were Caucasian and mixed ethnicity. According to the subscale cut-off scores of the CBCL, on average, children did not demonstrate problematic behaviors related to emotional reactivity, anxiety/depression, somatic complaints, withdrawal, attention, aggression, or sleep, at baseline. Children’s development was also on schedule upon enrolment, with the average ASQ-3 subscale scores above the cut-offs for communication, gross motor skills, fine motor, problem-solving, and personal social skills. Please refer to Table 4 for the child demographic characteristics from ATTACH^TM^ pilot study #6. A total of 10 caregivers and their children participated in ATTACH^TM^ study #7, which included the control group from ATTACH^TM^ pilot study #5.

#### 3.1.3. Comparisons between Samples

Before combining data from the two samples, one from the low-income family service agency (Pilots #4 and #5) and the other from the women’s shelter (Pilots #6 and #7), the samples were compared. Chi-square tests were performed for each of the caregivers’ sociodemographic variables to reveal no significant differences in age, primary language, the highest level of education, employment status, and whether parents were born in Canada. Caregivers who participated at the low-income family service agency were more likely to be Caucasian (x2 = 7.20, df = 1, *p* = 0.007) and single (x2 = 3.04, df = 1, *p* = 0.081) compared to those who participated at the women’s shelter. Chi-square tests were also performed for the children’s sociodemographic variables to reveal no significant differences in sex at birth. Children were more likely to be mixed ethnicity (x2 = 5.247, df = 1, *p* = 0.022) and speak English as a first language (x2 = 3.174, df = 1, *p* = 0.075) at the women’s shelter compared with those who participated at the low-income family service agency.

### 3.2. Outcomes

#### 3.2.1. RCTs (ATTACH^TM^ Pilot Studies #4 and #6)

Table 5 displays the results for the sample of participants who undertook the ATTACH^TM^ intervention during ATTACH^TM^ pilot RCTs #4 and #6. Caregivers in the intervention group demonstrated significantly improved parental RF regarding PRFQ interest and curiosity subscale scores compared with caregivers in the control group, post-intervention, and a medium effect size was observed. Caregivers in the intervention group demonstrated trends toward significantly improved parental RF regarding PRFQ pre-mentalizing subscale scores compared with caregivers in the control group, post-intervention, and a medium effect size was observed. There were no significant differences between the intervention group and control group on the PRFQ certainty about mental states subscale or FFMQ at post-intervention. Furthermore, there were no significant differences between the intervention group and control group, post-intervention, related to perceived social support on the SSE-Q and maternal executive function on the BRIEF-A.

Independent samples *t*-tests revealed that caregivers in the intervention groups reported that children demonstrated significantly higher ASQ-3 problem-solving skills subscale scores compared with children in the control group post-intervention, and a large effect size was observed. Caregivers in the intervention group reported that children demonstrated significantly improved ASQ-3 fine motor subscale scores compared with children in the control group post-intervention, and a large effect size was observed. We found no significant differences between the intervention group and control group on the ASQ-3 communication, personal social skills, or gross motor skills subscale scores, post-intervention. We also found no significant difference between the intervention group and control group on maternal perceptions of their children’s behavior on CBCL subscales for anxious/depressed behavior, attention problems, aggressive behavior, sleep problems, internalizing problems, or externalizing problems, post-intervention. No significant differences were observed for child-executive function on the BRIEF-P, post-intervention.

#### 3.2.2. QES (ATTACH^TM^ Pilots #5 and #7)

Table 6 displays the results for the sample of participants who undertook the ATTACH^TM^ intervention during pilot studies #5 and #7. For the combined sample, paired samples *t*-tests revealed that caregivers reported significantly improved parental RF on the PRFQ interest and curiosity subscale scores from baseline to post-intervention, and a medium-effect size was observed. Caregivers reported a significant increase in perceived social support from the person that they turn to most for social support on the SSE-Q from baseline to post-intervention, and a medium-effect size was observed. Furthermore, caregivers reported significantly improved maternal executive function on the BRIEF-A from baseline to post-intervention, and a large-effect size was observed. There were no significant changes between baseline and post-intervention related to the PRFQ pre-mentalizing and PRFQ certainty about mental states subscale scores.

Caregivers reported that children demonstrated a significant improvement in ASQ-3 communication scores from baseline to post-intervention and a significant improvement in ASQ-3 problem-solving scores from baseline to post-intervention, and a large-effect size was observed. Caregivers also reported a significant improvement in children’s ASQ-3 personal–social scores from baseline to post-intervention, and a medium-effect size was observed. Moreover, caregivers reported trends toward significant improvement in children’s ASQ-3 fine motor scores from baseline to post-intervention, and a small-effect size was observed. Significant decreases in CBCL externalizing problems, anxious/depressed behavior, sleep problems, attention problems, and aggressive behaviors were observed at post-assessment compared to baseline. No significant changes were observed between baseline and post-intervention regarding ASQ-3 gross motor scores and ASQ-3 fine motor scores, the CBCL internalizing problems total score, and child executive function on the BRIEF-P.

## 4. Discussion

This paper sought to examine the impacts of ATTACH™ on parental RF, perceived social support, and executive function, as well as children’s development, behavioral and sleep problems, and executive function. Parents (caregivers, who were mostly mothers) who took part in the Phase 2 RCTs, reported significantly improved parental RF and improved child development (i.e., problem-solving and fine motor skills) in the treatment group, compared with those in the control group, post-intervention. The caregivers in the Phase 2 QES reported significantly increased parental RF, executive function, and perceptions of social support, and children’s development (i.e., communication, problem-solving, and personal social skills) and decreased children’s behavioral problems (i.e., anxiety/depression, attention problems, aggressive behavior, and externalizing problems), and sleep problems from baseline to post-intervention. The results of Phase 2 confirmed many of the findings of Phase 1, including that ATTACH^TM^ improved parental RF [26,45] and child developmental outcomes [11]. These positive findings suggest that intervening during infancy and childhood may be essential to achieve long-lasting positive outcomes for both caregivers and children in families experiencing adversity and maltreatment.

### 4.1. Impact of ATTACH^TM^ on Parental RF and Mindfulness

Stress triggers a parent’s survival mode that over-activates the limbic system, impairing the critical parental reflective processes that are necessary to respond to children’s needs and provide them with a safe haven and secure base [111]. Thus, many interventions aim to improve outcomes for highly stressed populations at risk of maltreating children by enhancing parental RF [11,26,27,37,38,39,45] with outcomes often assessed via Fonagy’s RF Scale [26,30,37,45,112]. We utilized the PRFQ [93] that assesses three aspects of RF, including interest and curiosity in mental states, certainty about mental states, and pre-mentalizing modes, based on our previous research showing that Fonagy’s RF Scale and PRFQ are correlated [20]. Post-ATTACH™ intervention, caregivers’ scores improved on the interest and curiosity and pre-mentalizing PFRQ subscales, while FFMQ, focused on the related concept of mindfulness, failed to identify differences. These discrepant findings may be due to the closer ties between parental RF and interest and curiosity in mental states than the related concept of mindfulness [20,113]. Parents with lower levels of parental RF are less capable of demonstrating genuine curiosity about the subjective experience of their children, remaining in the pre-mentalizing mode [20,21,113]. Thus, when a parent is interested and curious about the mental states of their child, they may develop greater confidence in their knowledge of the child, resulting in more communication and involvement [20]. Mindfulness focused on internal representations and awareness at the moment [114] may not induce the same curiosity about children’s mental states.

### 4.2. Impact of ATTACH^TM^ on Parental Perceptions of Social Support

We found that ATTACH^TM^ enhanced perceived social support in the QES, consistent with literature indicating that higher social support likely has a positive impact on parenting skills, impacting parent–child relationship quality [87] and ultimately children’s development [115]. Increasing capacity for parental RF may positively affect parents’ broader relationships [13] as a newfound ability for insight into others’ thoughts and feelings may promote social cohesion and opportunities for reciprocity and mutual aid in the context of social support [116,117]. Such social support provides parents with increased resources to meet their infants’ and young children’s unrelenting emotional and physical care needs [21], potentially serving as a protective factor to children’s development, especially in families at risk of maltreatment. Our study may be the first to report the positive effects of parental RF intervention on caregivers’ perceptions of social support.

### 4.3. Impact of ATTACH^TM^ on Parental Executive Function

Our findings of the positive effects of ATTACH^TM^ on caregivers’ (mostly mothers’) executive function in the QES sample are consistent with other literature [58,118,119]. The ability to hold an infant in mind during periods of infant distress may facilitate a mother’s ability to handle her own emotions and behavior during such stressful situations [119], allocating more cognitive resources to their child [58]. Moreover, the ability of mothers to shift their behaviors during dyadic interactions may promote both flexibility in their caregiving behaviors and the ability to distinguish between their own and and their infants’ distress [119,120]. Thus, our study provides important evidence for the effect of parental RF intervention on maternal executive function and self-regulation.

### 4.4. Impact of ATTACH^TM^ on Child Development and Behavioural Problems

Building on our earlier observation that ATTACH™ positively impacted child development in the personal social domain [11], our Phase 2 identified additional impacts on problem-solving skills in both RCTs and QES and communication and fine motor skills in the QES. Thus, ATTACH^TM^ appears to enhance child development via parental RF. Positive impacts of parental RF-based intervention on externalizing behavior are consistent with the existing literature [77,121]. Our study also demonstrated significant decreases in sleep problems, attention problems, and aggressive behavior in the QES sample. An association has been observed between behavioral problems in childhood, particularly externalizing and internalizing behaviors [122], and an increased risk of poor lifelong developmental outcomes including psychopathology [123]. Thus, early intervention strategies, such as ATTACH™, may be crucial to preventing problem behaviors in childhood and beyond.

### 4.5. Impacts of ATTACH^TM^ on Child Executive Function

Extant empirical research has demonstrated an association between parental RF and executive function in mothers [58,119,124], which in turn, has been linked to children’s executive function [125,126]. Unexpectedly, ATTACH™ did not significantly affect children’s executive function. To thoroughly explore this association, future studies could consider delayingassessments or undertaking longer-term follow-up, as executive function in children may require more time to develop and be observable. Considering delayed post-intervention assessments as part of future data collection would be beneficial.

### 4.6. Strengths and Limitations

This study is characterized by several strengths, including a real-world examination of a promising intervention with a variety of ethnicities and ages of children involved. Limitations of the study include small sample sizes; however, we report medium to large effect sizes. Moreover, we worked with two different agencies that could have confounded findings. A further limitation of this study was the use of questionnaires completed by single respondents (i.e., caregivers only), which may introduce reporting bias.

### 4.7. Implications for Practice and/or Further Research

ATTACH^TM^ has the potential to be an effective intervention for widespread adoption to address risk for child maltreatment in agencies serving a range of high-risk families. Overall, we observed that the intervention contributed to significant increases in caregivers’ RF and children’s development in problem-solving and fine motor skills in the RCT. The ATTACH^TM^ intervention also contributed to a significant increase in parental executive function and perception of social support and children’s development (specifically, communication, social–emotional, and fine motor skills) and a significant decrease in children’s behavioral and sleep problems in the QES. Considering that childhood experiences of maltreatment have lasting and complex effects, enhancing parental RF may be a critical component of interventions aimed at promoting child development and behavior in high-risk families.

## 5. Conclusions

The ATTACH^TM^ intervention is a promising program that stands to optimize parenting, child development, and behavior in high-risk families exposed to adversity (e.g., family violence, parental depression, low income) and protect children against maltreatment. Our findings indicate that parental RF supports multiple relational, interpersonal, intrinsic, and behavioral capacities that are broadly protective throughout life. ATTACH™ can be implemented in a variety of social service agencies to develop these capacities and create long-lasting change in families at risk of child maltreatment, with the potential for significant intergenerational impacts.

## Figures and Tables

**Table 1 ijerph-20-03078-t001:** Caregiver Demographics for ATTACH^TM^ Pilot Study #4 (Baseline).

	*n*	Percent or Mean (SD)
Age (Years)	14	30.8 (7.55)
Relationship to Child		
Mother	13	92.9%
Grandmother	1	7.2%
Ethnicity		
African	1	7.2%
Caucasian	9	64.3%
Hispanic	1	7.2%
Indigenous	1	7.2%
Mixed ethnicity	2	14.3%
Primary Language		
English	11	78.6%
Other	3	21.4%
Born in Canada		
No	4	28.6%
Yes	10	71.4%
Education		
Some high school	4	28.6%
High school diploma	6	42.9%
Some post-secondary	4	28.6%
Marital Status		
Partnered	4	28.6%
Single	10	71.4%
Employment Status		
Full-time	8	57.1%
Maternity leave	1	7.2%
Unemployed/not in the workforce	4	28.6%
Missing	1	7.2%

**Table 2 ijerph-20-03078-t002:** Child Demographics for Pilot Study #4 (Baseline).

	*n*	Percent or Mean (SD)
Age (Months)	14	25.0 (6.60)
Gender		
Male	8	57.1%
Female	6	42.9%
Ethnicity		
African	2	14.3%
Caucasian	9	64.3%
Indigenous	1	7.2%
Mixed ethnicity	2	14.3%
Primary Language		
English	14	100%

**Table 3 ijerph-20-03078-t003:** Caregiver Demographics for Pilot Study #6 (Baseline).

Variable	*n*	Percent or Mean (SD)
Age (Years)	20	31.6 (4.96)
Relationship to Child		
Caregiver	20	100%
Ethnicity		
Afghani	1	5.00%
African	2	10.0%
Asian	2	10.0%
Caucasian	5	25.0%
Hispanic	1	5.00%
Indigenous	7	35.0%
Mixed ethnicity	2	10.0%
Primary Language		
English	17	85.0%
Other	3	15.0%
Born in Canada		
No	7	35.0%
Yes	13	65.0%
Education		
Less than high school	2	10.0%
Some high school	4	20.0%
High school diploma	6	30.0%
Some college	8	40.0%
Marital Status		
Partnered	1	5.00%
Single	19	95.0%
Employment Status		
Part-time	4	20.0%
Full-time student	1	5.00%
Unemployed/not in workforce	14	70.0%
Other	1	5.00%

**Table 4 ijerph-20-03078-t004:** Child Demographics for Pilot Study #6 (Baseline).

	*n*	Percent or Mean (SD)
Age (Months)	20	30.8 (17.5)
Gender		
Male	9	45.0%
Female	11	55.0%
Ethnicity		
Afghani	1	5.00%
African	2	10.0%
Asian	2	10.0%
Caucasian	5	25.0%
Hispanic	1	5.00%
Indigenous	2	10.0%
Mixed ethnicity	5	25.0%
Pacific Island	2	1.00%
Primary Language		
English	17	85.0%
Non-English	3	15.0%

**Table 5 ijerph-20-03078-t005:** Between-Group Comparisons of Outcome Measures for Randomized Controlled Trials at Post-Intervention (Pilots #4 and #6 combined).

	*n*	Control Group	Intervention Group	t	*p*	d
		M (SD)	M (SD)			
Parental Outcome Measures						
PRFQ (Interest and Curiosity in Mental States Subscale)	31	5.14	5.71	1.81	0.020 *	0.65
		1.07	0.63			
PRFQ (Pre-Mentalizing)	31	2.53	2.08	−1.27	0.053	0.46
		1.08	0.88			
PRFQ (Certainty of Mental States)	31	4.28	4.22	−0.13	0.224	0.05
		1.38	1.07			
FFMQ (Observing Subscale)	31	25.69	26.07	0.18	0.215	0.06
		5.74	6.31			
FFMQ (Describing Subscale)	31	28.38	26.47	−0.87	0.100	0.31
		6.38	5.79			
FFMQ (Acting with Awareness Subscale)	31	29.25	26	−1.49	0.074	0.54
		7.05	4.8			
FFMQ (Non-Judging of Inner Experiences Subscale)	31	25.81	25.8	−0.01	0.249	0.00
		7.11	6.52			
FFMQ (Non-Reactivity to Inner Experiences Subscale)	31	20.81	21.07	0.07	0.224	0.05
		6.3	3.71			
SSE-Q (Total Score)	29	33.73	34.07	0.07	0.237	0.03
		11.63	15.23			
BRIEF-A (Global Executive Composite Score)	31	38.19	47.13	0.76	0.114	0.27
		36.74	28.29			
Child Outcome Measures						
ASQ-3 (Communication Skills Subscale)	30	43.44	45.71	0.41	0.172	0.15
		13	17.64			
ASQ-3 (Problem-Solving Skills Subscale)	30	41.25	51.79	2.04	0.025 *	0.76
		17.17	9.52			
ASQ-3 (Personal–Social Skills Subscale)	30	0.07	0.05	0.3	0.192	0.44
		0.05	0.04			
ASQ-3 (Fine Motor Skills Subscale)	30	35.94	48.93	2.2	0.018 *	0.81
		16.66	15.46			
ASQ-3 (Gross Motor Skills Subscale)	30	35.94	48.93	−0.26	0.130	0.81
		16.65	15.46			
CBCL (Anxious/Depressed Subscale)	31	2.69	2.73	−0.05	0.479	0.21
		1.99	2.71			
CBCL (Sleep Problems Subscale)	31	1.81	1.20	0.86	0.199	0.30
		2.37	1.47			
CBCL (Attention Problems Subscale)	31	2.31	2.33	−0.03	0.488	0.01
		2.47	1.35			
CBCL (Aggressive Behaviour Subscale)	31	6.88	7.53	−0.29	0.385	0.01
		7.00	5.30			
CBCL (Externalizing Problems Total Score)	31	9.19	9.87	0.25	0.202	0.09
		9.14	5.78			
CBCL (Internalizing Problems Total Score)	31	8.31	10.67	0.93	0.091	0.33
		7.52	6.57			
BRIEF-P (Total Score)	31	27.75	30.07	0.34	0.185	0.12
		21.8	15.92			

** p* < 0.05 one-tailed. Note: ASQ-3 = Ages and Stages Questionnaire, Third Edition; CBCL = Child Behaviour Checklist; BRIEF-P = Behaviour Rating Inventory of Executive Function—Preschool Version; PRFQ = Parental Reflective Functioning Questionnaire; SSE-Q = Social Support Effectiveness Questionnaire; BRIEF-A = Behaviour Rating Inventory of Executive Function—Adult Version; FFMQ = Five Facets of Mindfulness Questionnaire.

**Table 6 ijerph-20-03078-t006:** Within-Group Comparisons of Outcome Measures for Quasi-Experimental Studies (Pilots #5 and #7 combined).

	*n*	Baseline	Post-Intervention	t	*p*	d
		M (SD)	M (SD)			
Parental Outcome Measures						
PRFQ (Interest and Curiosity in Mental States Subscale)	13	5.19	5.9	−2.04	0.032 *	0.57
		1.18	0.8			
PRFQ (Pre-Mentalizing)	13	2.37	2.15	0.44	0.167	0.12
		0.94	1.33			
PRFQ (Certainty of Mental States)	13	4.42	4.47	−0.16	0.218	0.04
		1.35	1.69			
FFMQ (Observing Subscale)	13	25.77	23.46	0.92	0.094	0.26
		5.95	9.55			
FFMQ (Describing Subscale)	13	27.69	27.08	0.36	0.182	0.1
		6.21	5.68			
FFMQ (Acting with Awareness Subscale)	13	30.15	29.85	0.19	0.214	0.05
		6.91	8.3			
FFMQ (Non-Judging of Inner Experiences Subscale)	13	26.85	29.08	−1.07	0.249	0.3
		6.32	7.19			
FFMQ (Non-Reactivity to Inner Experiences Subscale)	13	20.85	18.15	1.04	0.076	0.29
		7.01	7.82			
		5.06	4.03			
SSE-Q (Total Score)	13	47.08	55.54	1.95	0.037 *	0.54
		18.43	11.66			
BRIEF-A (Global Executive Composite Score)	13	33.1	22.6	2.31	0.020 *	0.64
		32.7	29.8			
Child Outcome Measures						
ASQ-3 (Communication Skills Subscale)	14	45.71	52.14	−2.86	0.006 *	0.76
		11.91	12.35			
ASQ-3 (Problem-Solving Skills Subscale)	14	42.5	53.21	−2.84	0.007 *	0.76
		18.05	12.49			
ASQ-3 (Personal–Social Skills Subscale)	14	50.71	55	−1.79	0.048 *	0.48
		8.51	9.2			
ASQ-3 (Fine Motor Skills Subscale)	14	37.5	43.21	−1.23	0.060	0.33
		17.29	15.89			
ASQ-3 (Gross Motor Skills Subscale)	14	51.79	51.07	0.2	0.212	0.05
		12.5	9.44			
CBCL (Anxious/Depressed Subscale)	13	2.38	1.46	2.22	0.023 *	0.62
		1.80	1.45			
CBCL (Sleep Problems Subscale)	13	1.92	0.92	1.88	0.042 *	0.52
		2.46	1.32			
CBCL (Attention Problems Subscale)	13	2.46	1.46	2.66	0.010 *	0.74
		2.60	1.61			
CBCL (Aggressive Behaviour Subscale)	13	7.08	4.62	1.81	0.048 *	0.5
		7.24	4.35			
CBCL (Externalizing Problems Total Score)	13	9.54	6.08	2.31	0.020 *	0.64
		9.45	5.48			
CBCL (Internalizing Problems Total Score)	13	7.54	10.67	0.93	0.147	0.26
		7.52	6.57			
BRIEF-P (Total Score)	13	27.38	23.54	0.73	0.120	0.2
		22.24	23.76			

** p* < 0.05 (one-tailed). Note: ASQ-3 = Ages and Stages Questionnaire, Third Edition; CBCL = Child Behaviour Checklist; BRIEF-P = Behaviour Rating Inventory of Executive Function—Preschool Version; PRFQ = Parental Reflective Functioning Questionnaire; SSE-Q = Social Support Effectiveness Questionnaire; BRIEF-A = Behaviour Rating Inventory of Executive Function—Adult Version; FFMQ = Five Facets of Mindfulness Questionnaire.

## Data Availability

Requests to access the data supporting the findings can be directed to N.L. at nicole.letourneau@ucalgary.ca.

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
