# Peer review of "Impacts of the Attachment and Child Health (ATTACH^TM^) Parenting Program on Mothers and Their Children at Risk of Maltreatment: Phase 2 Results"

_ijerph, 2023, doi:10.3390/ijerph20043078_

Round 1

Reviewer 1 Report

The study investigates the impacts of the Attachment and Child Health Parenting Program on Mothers and their Children at Risk of Maltreatment: Phase 2 Results. What makes this study interesting is the use of an interesting program to increase mothers’ capacity to develop skills that will allow diminish the possibility of maltreatment. I think that the findings of this study well fit into the Journal’s scope and should be published. However, there are some issues regarding the presentation of methods and, in particular the discussion which offers a minor revision of the paper. I am optimistic that such a revision is possible.
The introduction provides a good impression of the state of knowledge in the field.
Method
The method section provides a clear picture of the procedures and measures used. The design of the study is well-done. I would recommend adding some arguments for deciding the choice of the measures and clearly stating that all the data was collected using indirect measures, and mothers were the single respondents.
The section on statistical analyses is simple but clear. In the discussion, there should be a more profound argument about the limitations of the study and future work. It is not only a problem of a small sample, I think an important limitation is the use of the mother as a single informant. It is the mother’s representation of the child’s behavior, and it would be very important to acknowledge the importance of other informants.

The paper will be important to the dissemination of the program to other samples, and the continuation of using evidence base programs to work with families and children.

Author Response

Response to Reviewer 1 Comments

Point 1: The study investigates the impacts of the Attachment and Child Health Parenting Program on Mothers and their Children at Risk of Maltreatment: Phase 2 Results. What makes this study interesting is the use of an interesting program to increase mothers’ capacity to develop skills that will allow diminish the possibility of maltreatment. I think that the findings of this study well fit into the Journal’s scope and should be published. However, there are some issues regarding the presentation of methods and, in particular the discussion which offers a minor revision of the paper. I am optimistic that such a revision is possible.

The introduction provides a good impression of the state of knowledge in the field.

Method

The method section provides a clear picture of the procedures and measures used. The design of the study is well-done. I would recommend adding some arguments for deciding the choice of the measures and clearly stating that all the data was collected using indirect measures, and mothers were the single respondents.

Response 1: Thank you for your great feedback. We described the choice of the measures in lines 236 to 238 as follows. “We selected measures to align with the recommendations of the Harvard Center of the Developing Child’s Frontier of Innovation program [92].”

Also, we added that the questionnaires were completed by single respondents in lines 535-536 as follows. “A further limitation of this study was the use of questionnaires completed by single respondents (i.e., mothers only), which may introduce self-report bias.”

Point 2: The section on statistical analyses is simple but clear. In the discussion, there should be a more profound argument about the limitations of the study and future work. It is not only a problem of a small sample, I think an important limitation is the use of the mother as a single informant. It is the mother’s representation of the child’s behavior, and it would be very important to acknowledge the importance of other informants.

The paper will be important to the dissemination of the program to other samples, and the continuation of using evidence base programs to work with families and children.

Response 2: We discussed the limitations of the study in lines 535-536 as above. We hope that you find the changes to be satisfactory.

Reviewer 2 Report

Comments included in attached document

Author Response

Response to Reviewer 2 Comments

Point 1: Overall

This paper examines the impact of an intervention aimed at improving parental reflective function on parent and child outcomes. It is an interesting paper with the potential to contribute valuable information to the field of child maltreatment prevention. However, there are some issues in terms of the discussion of results, wherein the authors report finding significant associations for outcomes, when many of these outcomes were significant in one sample but not the other. These differences are outlined in the results and Tables, but not addressed when describing the findings or their implications in the Discussion.

Abstract

Pg 1, lines 26-27: “Parents with their children aged 0-5 years (N=45), experiencing adversity received the 10-12-week ATTACHTM intervention.” Clarify whether it’s the parent, child, or both who are experiencing adversity. Examples of suggested revisions “Parents experiencing adversity, along with their children aged 0-5 years, received…” or “Parents and their children (age 0-5 years) who were experiencing adversity received….”

Response 1: Thank you for your great feedback. We have made the revisions in lines 27-29 as follows. “Parents experiencing adversity, along with their children aged 0-5 years (N=45), received the 10-12-week ATTACHTM intervention.”

Point 2: Introduction

Pg 1-2, lines 43-46: “In contrast, secure parent -child attachment, underpinned by parental reflective function (RF), defined as the parent’s ability to think about and identify thoughts, feelings and mental states in themselves and in their children, is linked to positive health and developmental outcomes in children [12-15]. RF operationalizes mentalizing…” Recommend breaking the first sentence up: “In contrast, secure parent - child attachment, underpinned by parental reflective function (RF), is linked to positive health and developmental outcomes in children [12-15]. RF is defined as the parent’s ability to think about and identify thoughts, feelings and mental states in themselves and

in their children. RF operationalizes mentalizing…”

Response 2: Thank you. We revised it in lines 43-49 as follows. “In contrast, secure parent-child attachment, underpinned by parental reflective function (RF), is linked to positive health and developmental outcomes in children [9-12]. Parental RF is defined as the parent’s (or primary caregivers’) ability to think about and identify thoughts, feelings, and mental states in themselves and in their children [13, 14]. According to leading theorist Peter Fonagy, RF operationalizes mentalizing which involves attending to mental states in oneself and others and interpreting behaviour accordingly [15, 16].”

Point 3: Pg 2, lines 59-61: “...including those affected by parental histories of early trauma or adversity, and current drug and alcohol abuse, psychopathology, and dysfunctional parenting.” Suggested revision: “...including those affected by parental histories of early trauma or adversity, current drug and alcohol abuse, psychopathology, and/or dysfunctional parenting”

Response 3: We revised it in lines 66-68 as follows: “including those affected by parental histories of early trauma or adversity, current drug and alcohol abuse, psychopathology, and/or dysfunctional parenting [3, 7, 31-33].”

Point 4: Pg 2, line 62: “socio-economic risks associated with low-income” Suggested revision: “risks associated with low income”

Response 4: We made the revisions in line 69 as follows. “socio-economic risks”.

Point 5: Pg 2, lines 58-76: Recommend switching the order of these two paragraphs (“The ability of parents…” and “Parents who are highly….” to improve flow/organization

Response 5: Thank you. We have switched the order of those paragraphs in lines 58-72 as follows.               “Parents who are highly reflective interpret their child’s behaviour by considering their own mental and those of their children as well as interactions between the each other’s mental states [10, 11, 13] while those with low RF tend to be unaware of their own internal experiences and/or their children’s mental states [13, 20]. Parental RF has been linked to parent-child interaction quality, with higher RF associated with sensitive parenting and lower RF associated with disruptive parenting quality [11, 26-30].

The ability of parents to be reflective during interactions with their children is often observed to be suboptimal in populations at risk for maltreatment, including those affected by parental histories of early trauma or adversity, current drug and alcohol abuse, psychopathology, and/or dysfunctional parenting [3, 7, 31-33]. High-risk families are also likely to be exposed to intersecting socio-economic risks including low maternal education, single parenthood, young age at conception, and minority status [34]. Moderate and higher RF ability in mothers predicts secure infant attachment, in both low [10, 20, 35] and high-risk [32, 36] samples.”

Point 6: Pg 2, lines 79-81: “Also, parental RF interventions improved RF for mothers with addiction [28, 29, 31-33, 44], imprisonment [45, 80 46], and experiencing depression, family violence, poverty and addiction” Suggested revision: “Also, parental RF interventions improved RF for mothers experiencing addiction [28, 29, 31-33, 44], imprisonment [45, 80 46], depression, family violence, and/or poverty”

Response 6: We revised it in lines 77-78 as follows. “Parental RF interventions also improved RF for mothers experiencing addiction [28, 38-41], imprisonment [43-45], depression, family violence, and/or poverty [11, 27, 46].”

Point 7: Pg 2, line 85: “As a social support system, perceived social support is defined as…”

Recommend removing “As a social support system”

Response 7: We revised it in line 82 and removed “As a social support system.”

Point 8: Pg 2, lines 87-94: Recommend switching the order of these two sentences (“While

parental RF…” and “A parent’s perception” to improve flow/organization

Response 8: We switched the order of those sentences in lines 145-149 (and edited it as per reviewer #3) as follows. “The more stress parents experience in their close relationships (e.g., having no support or less support) the more challenging it may be to focus on their children’s mental states and behaviours [86, 88]. Parental RF may impact interrelatedness with others in the social network, which may increase perception of the quality and quantity of social support [86].”

Point 9: Pg 3, line 105: Suggest changing “It is undeniable” to “Research shows” or something

along those lines.

Response 9: Thank you. We removed the word “It is undeniable”.

Point 10: Pg 3, line 108: Suggest changing “heterogeneous participation” to “non-representative samples” or “inconsistent sampling” or something like that, because heterogenous participation could just mean a diverse group of participants, which wouldn’t be a limitation.

Response 10: We made the revisions in line 101-104 as follows. “However, it is unclear to what extent these improvements are generalizable to parents and translate into positive changes in RF or children’s behavior [51]. To our knowledge, no studies have examined the impact of interventions focused on parental RF on parents’ and children’s executive function.”

Point 11: Pg 3, lines 112-115: “It has been demonstrated that mothers who had a better working

memory were more interested in their child's feelings and curiosity, while mothers with better visual working memory and set-shifting capacities also showed higher levels of interest in and curiosity about their children's thoughts and feelings” The “while” indicates that the second part of the sentence contrasts with the first, but both parts of the sentence appear to both support the same general finding (i.e., that better working memory is associated with greater interest/curiosity in child’s feelings). Suggest rephrasing to clarify.

Response 11: Thank you. We revised it in lines 84-92 as follows. “Compared to mothers with less optimal working memory, those with better working memory are more capable of managing their own emotions and behavioral responses to their distressed infants [51-55], display more interest in their child's feelings and curiosity [56], and more positive behaviours towards their children in a frustration-based task [57, 58]. Mothers with better set-shifting capacities show higher levels of interest and curiosity in their children's thoughts and feelings, and adjust their behaviors during dyadic interactions to accommodate their children, consistent with higher parental RF [59].”

Point 12: Pg 3, lines 124-125: “Considering how relevant executive function may be to parental

RF, it is paramount to support at-risk parents and children, as well as fill the gap in the literature.” Is this sentence referring specifically to supporting parents in their executive function? Or is it saying that because executive function can impact RF, parents and children should be supported in general? Recommend revising to clarify the link between the two ideas in this sentence.

Response 12: We have revised it in lines 101-104 as follows. “However, it is unclear to what extent these improvements are generalizable to parents and translate into positive changes in RF or children’s behavior [51]. To our knowledge, no studies have examined the impact of interventions focused on parental RF on parents’ and children’s executive function.”

Point 13: Pg 3, lines 129-132: “For example, mothers with verbal working memory, as an aspect of executive function, exhibited greater negative behaviour towards their children in relation to a frustration-based task” Greater negative behavior compared to what group? Mothers with other types of working memory?

Response 13: We revised it in lines 85-92 as follows. “Compared to mothers with less optimal working memory, those with better working memory are more capable of managing their own emotions and behavioral responses to their distressed infants [51-55], display more interest in their child's feelings and curiosity [56], and more positive behaviours towards their children in a frustration-based task [57, 58]. Mothers with better set-shifting capacities show higher levels of interest and curiosity in their children's thoughts and feelings, and adjust their behaviors during dyadic interactions to accommodate their children, consistent with higher parental RF [59].”

Point 14: Pg 3, line 140: Suggest changing “would” to “may”

Response 14: Thank you. We made extensive edits in response to reviewer #3 comments and revised that sentence as follows in lines 101-104. “However, it is unclear to what extent these improvements are generalizable to parents and translate into positive changes in RF or children’s behavior [51]. To our knowledge, no studies have examined the impact of interventions focused on parental RF on parents’ and children’s executive function.”

Point 15: Pg 3, lines 144-146: “However, children's development has been demonstrated to be

less substantially impacted by parental RF interventions, with significant differences observed mainly on the personal social domain of child development” Recommend revising this sentence for clarity. Is it saying parental RF interventions have mostly shown impacts on personal/social domains of child development, but not other domains?

Response 15: Thank you. We revised it in lines 107-109 as follows: “Parental RF interventions have demonstrated impacts on personal-social domains of child development but not other domains such as communication, problem-solving, and gross and fine motor skills [11].”

Point 16: Pg 4, lines 151-152: I don’t think mixed results would be a limitation of the study itself.

Maybe another way to phrase it would be changing “The limitations of those studies include small sample sizes [14, 77], and reporting limitations such as a lack of blind assessments [77] and mixed results [78, 79]” to “These studies had mixed findings and had limitations such as small sample sizes [14, 77], and lack of blind assessments [77]”

Response 16: We revised it in lines 113-114 as follows. “These studies had limitations such as small sample sizes [11, 66], and lack of blind assessments [66].”

Point 17: Pg 4, lines 158-160:“A significant association was found between mothers’ more

optimal RF and improved social-emotional skills, less socio-emotional and behaviour problems in children” Incomplete sentence as written. Suggested revision “A significant association was found between mothers’ more optimal RF and improved social-emotional skills as well as fewer socio-emotional and behavioural problems in children.”

Response 17: We made the revisions in lines 118-119 as follows. “A significant association was found between mothers’ higher RF and improved social-emotional skills as well as fewer socio-emotional and behavioral problems in children [69-74].”

Point 18: Pg 4, lines 160-164: “In contrast, children of parents with low parental RF are more likely to display emotion dysregulation, anxiety disorders, internalizing, and externalizing

behaviors [74], and suboptimal parental RF is linked to higher anxiety in children [87],

Attention-Deficit Hyperactivity Disorder (ADHD) [88], and lower social–emotional

competencies in children [86, 89]” Suggest combining these sentences because some of

the information is repeated. “In contrast, children of parents with low parental RF are more likely to display emotion dysregulation, anxiety disorders, Attention-Deficit Hyperactivity Disorder (ADHD) [88], internalizing and externalizing behaviors [74], and lower social–emotional competencies [86, 89]”

Response 18: We have revised the sentences in lines 120-123 as follows. “In contrast, children of parents with low parental RF are more likely to display emotion dysregulation, anxiety disorders, Attention-Deficit Hyperactivity Disorder (ADHD) [75], internalizing and externalizing behaviors [76], and lower social–emotional competencies [74, 77].”

Point 19: Pg 4, 166-168: “Another RF-based parenting intervention for foster parents, Family

Minds, reported no significant changes in children's emotional and behavioral difficulties specifically internalizing behaviors, post-intervention [91, 92]” Needs additional punctuation. Suggested revision: “Another RF-based parenting intervention for foster parents, Family Minds, reported no significant changes in children's emotional and behavioral difficulties– specifically internalizing behaviors– post-intervention [91, 92]” (Could also use parentheses if preferred)

Response 19: We revised it in lines 125-127 as follows. “Another RF-based parenting intervention for foster parents, Family Minds, reported no significant changes in children's internalizing behaviors (e.g. anxiety, social withdrawal), post-intervention [79, 80].”

Point 20: Pg 4, 172-187: The paragraph about sleeping disorders seemed out of place since the

header indicated the paragraph would be about behavioral problems. (I’m not wellversed in sleep research though, so maybe sleep problems are characterized as behavioral problems in the field).

Response 20: Thank you. We have changed the title of the heading in line 116 as follows. “1.4. Parental RF and Children’s Behavioural and Sleep Problems”. Also, we added it in the abstract in line 31.

Point 21: Pg 4, line 193: Change “design” to “designed”

Response 21: Thank you. It has been revised it in line 156.

Point 22: Pg 5, line 207: Recommend identifying the longstanding outcomes of interest (the author mentioned them early on in the intro, but it would be helpful to reiterate all outcomes of

interest here)

Response 22: We revised it in lines 167-171 as follows. “The current report aims to build on findings to date from Phase 1 and Phase 2 pilots, this time to determine the impact of ATTACHTM on outcomes of long-standing interest including parental RF and child development, as well as new outcomes examining parents’ executive function and perceived social support and children’s behavioural problems, sleep, and executive function.”

Point 23: Pg 5, line 208: “executive function, perceived social support” Change comma to “and”

since both of these refer to parent outcomes (similar to what you did for the two child outcomes)

Response 23: We revised it in line 170.

Point 24: Pg 5, lines 211-213: “...as well as improved parental perceptions of social support, and

parental executive functioning, and reduced behavioural problems in children and improved executive function.” Suggested revision: “...as well as improved executive functioning and perceptions of social support in parents and reduced behavioural problems and improved executive function in children”

Response 24: Thank you. We revised it in lines 169-171 as follows. “as well as new outcomes examining parents’ executive function and perceived social support and children’s behavioural problems, sleep, and executive function.”

Point 25: Methods

Pg 5, lines 215-217: “Guided by the Innovate, Develop, Evaluate, Adapt, Scale (IDEAS;

[7, 103]) Impact Framework that emphasizes rapid cycling trial methods, Phase 1, now completed involved pilots #1 to #3” Suggested revision: “Guided by the Innovate, Develop, Evaluate, Adapt, Scale (IDEAS; [7, 103]) Impact Framework that emphasizes rapid cycling trial methods, Phase 1 (now completed) involved pilots #1 to #3.”

Response 25: We revised it in line 177-179 as follows. “Guided by the Innovate, Develop, Evaluate, Adapt, Scale (IDEAS; [3, 7, 93]) Impact Framework that emphasizes rapid cycling trial methods, Phase 1 (now completed) involved pilots #1 to #3/”

Point 26: Pg 5, line 236: Change “lasts” to “last”

Response 26: We revised it in line 199.

Point 27: Pg 5, line 240: I think it could be helpful to provide an example of a hypothetical and a

real-life scenario used in the intervention. It could help clarify how a real-life scenario is different from a hypothetical one in the context of an observed clinical intervention.

Response 27: Thank you for your comment. We have provided some examples of a hypothetical situation and a real-life situation in lines 203-207 as follows. “For example, the hypothetical situation from session #1 asks parents to consider family members’ thoughts and feelings during a mealtime when the child drops their food on the floor. Real-life situations derive from parents’ stressful experiences over the past week, in which parents are asked to think about the thoughts and feelings of everyone involved.”

Point 28: Pg 6, line 270-Pg 7, line 332: If possible, I would try to provide the same info about validity/reliability for each measure. For the first one, only convergent validity is described; for the second one, several different measures are described; for the third one, only internal consistency is described, etc.

Response 28: Thank you. We have now provided some more details about validity/reliability for each measure as follows:

- Lines 247-249: “The PRFQ pre-mentalizing, certainty about mental states and interest and curiosity subscales demonstrated excellent construct validity, internal consistency, and reliability [21].”

- Lines 255-257: “Studies have demonstrated excellent construct validity [96], favorable discriminant validity and convergent validity [96, 97], and excellent internal consistency as well as incremental validity [96] of the FFMQ.”

- Lines 264-265: “The BRIEF-A has demonstrated excellent internal consistency and convergent validity [99].”

- Lines 275-276: “The SSE-Q is well validated, and exhibits demonstrated excellent internal consistency (α = .95) [101].”

- Lines 284-284: “The ASQ-3 has demonstrated excellent validity (between 0.82 - 0.88), sensitivity of 86%, and specificity of 85% [104].”

- Lines 294-296: “The CBCL has excellent convergent validity [105], and displays high internal consistency (α = 0.87 and 0.89 for the internalizing and externalizing problem scales, respectively [106, 107].”

- Lines 303-305: “The BRIEF-P has demonstrated excellent internal consistency [109], test-retest reliability [110], and content validity [111].”

Point 29: Pg 7, lines 310-311: “The questionnaire is a series of 21 parent-completed questionnaires” Should this be 21 parent-completed items, or is it a questionnaire made up of 21 different questionnaires?

Response 29: Thank you. We clarified it in lines 280-281 as follows.The questionnaire is a series of parent-completed, age specific, 21 different questionnaires.”

Point 30: Results

Overall: There are so many results for this study that this section ends up being a little challenging for the reader. All of the results are provided in the tables, so I think it could be beneficial to streamline the reporting of results in the text.

Response 30: We deleted all the redundant information from the text. Please see lines 322-442.

Point 31: Pg 8, line 354: “moderate average”Recommend removing one of these words

Response 31: Thank you. We revised it in line 327 as follows. “moderate level.”

Point 32: Pg 9, line 383: “moderate average” Recommend removing one of these words

Response 32: We revised it in line 350 as follows. “moderate level.”

Point 33: Pg 12, Table 5: Since the 0.053 was approaching statistical significance but was past the cutoff, it’s inaccurate to label it with the asterisk (which is described in the key as being <0.05)

Response 33: Thank you. We removed the asterisk in Table 5 for the p-value of 0.053.

Point 34: Pg 16, Table 6: It’s inaccurate to label 0.06 with the asterisk (which is described in the key as being <0.05)

Response 34: We removed the asterisk in Table 5 for the p-value of 0.060.

Point 35: Discussion

Pg 16, lines 552-553: “We found that ATTACHTM enhanced parental perceived social support in accordance with literature indicating the significant role of parental RF in parenting” Indicate that this was significant in one sample but not the other.

Response 35: We revised it in lines 488 as follows. “We found that ATTACHTM enhanced perceived social support in the QES.”

Point 36: Pg 16, lines 558-560: “While parents may encounter difficulties in their close relationship and in the development of RF skills, children, especially infants are dependent on their

parents for emotional and physical care on….” Needs additional punctuation. Suggested

revision: “While parents may encounter difficulties in their close relationship and in the development of RF skills, children– especially infants– are dependent on their parents for emotional and physical care on….”

Response 36: We revised it in lines 491-494 as follows. “Increasing capacity for parental RF may positively affect caregivers’ broader relationships [13] as newfound ability for insight into others’ thoughts and feelings may promote social cohesion and opportunities for reciprocity and mutual aid in the context of social support [117, 118].”

Point 37: Pg 16, lines 558-559: “Our findings of the positive effects of ATTACHTM on maternal

executive function are consistent with other literature [61, 64, 67]” Indicate that this was significant in one sample but not the other

Response 37: Thank you. We revised it in lines 501-502 as follows. “Our findings of the positive effects of ATTACHTM on caregivers’ (mostly mothers’) executive function in the QES sample are consistent with other literature.”

Point 38: Pg 17, lines 585-586: “Our study also demonstrated significant decreases in sleep

problems, attention problems, and aggressive behavior.” Indicate that these were significant in one sample but not the other

Response 38: Thank you. We revised it in lines 516-517 as follows. “Our study also demonstrated significant decreases in sleep problems, attention problems, and aggressive behavior in the QES sample.”

Point 39: Pg 17, lines 594-595: “Unexpectedly, significant changes in children’s executive function were not significant.” This sentence is unclear– recommend changing “significant changes” to “changes”

Response 39: We revised it in lines 525-526 as follows. “Unexpectedly, ATTACH™ did not significantly affect children’s executive function.”

Point 40: Pg 17, lines 600-603: Additional detail and discussion regarding the study’s strengths and limitations would be beneficial to include.

Response 40: Thank you. We added limitations in lines 535-536 as follows. “A further limitation of this study was the use of questionnaires completed by single respondents (i.e., mothers only), which may introduce parent-report bias.’

Point 41: Pg 17, lines 606-611: “Overall, we observed that the intervention contributed to significant increases in caregiver’s RF, executive functioning, and perception of social support. The ATTACHTM intervention also contributed to a significant increase in children’s development (specifically, communication, social-emotional, and fine motor skills) and a significant decrease in children’s behavioural problems.” Significant findings were not consistent across the board for both samples.

Response 41: Thank you. We revised it in lines 539-545 as follows. “Overall, we observed that the intervention contributed to significant increases in caregivers’ RF and children’s development in problem solving and fine motor skills in the RCT. The ATTACHTM intervention also contributed to a significant increase in parental executive function and perception of social support, and children’s development (specifically, communication, social-emotional, and fine motor skills) and a significant decrease in children’s behavioral and sleep problems in the QES.”

Point 42: Conclusion

No comments

Response 42: Thank you.

Point 43: Other comments

Check reference list to ensure they are in the correct format (MDPI | Reference List and Citations Style Guide).

Response 43: Thank you. We checked the reference list for correct formatting according to the MDPI Reference List and Citations Style Guide. We hope that you find the changes to be satisfactory.

Reviewer 3 Report

The manuscript presents the results of a valuable project, which deals with the problem of possible negative developmental outcomes in children at risk for maltreatment. Building on completed Phase 1 pilot data, Phase 2 examined several outcomes including parental reflective function and child development, as well as parental perceived social support and executive function, and children’s behaviour and executive function.

Although the manuscript is well written, in my opinion some changes could be made to make it clearer and easier to read.

The Introduction is too long. It needs to be shortened and more focused on specific goal of this research which is to determine the impact of ATTACHTM on parental RF and children’s development outcomes of interest.

Reflective function as s main construct of interest should be better explained and elaborated within some theory. A lot is written about RF and its relations to other constructs included in this study, but the theoretical frame is missing or should be more emphasized.

e.g. Lines 48 and 49 - RF differs from related terms such as mindblindedness, mindreading, theory of mind, metacognition, mindfulness, empathy and emotional intelligence. How? In what way? From the given definition, these constructs are quite similar.

I propose to the authors to shorten the introduction, in a way that they first give some theoretical frame for the examined constructs, short overview of the programs for interventions and their own program. Authors could shorten the description of research results for every outcome. There is a lot of text and information in the Introduction, which makes the text hard to follow.

Participant and measures are adequately described, and statistical analyses are appropriate.

In the Results section, Tables 1-4 can be removed from the manuscript. Demographic information of the parents and children are described in the text and there is no need to show it again in the table. If authors think this information are vital, they can be presented at the end of the manuscript or in the supplement material.

Instead of these tables, it would be helpful to have one table with the important information about the design – the overview of the intervention with information on the participant at each phase and pilot studies, and outcomes of interest.

Please do not mark results with p = .053 as significant (lines 427 and Table 5).

Discussion – please start this section with brief introduction, not with the results. The overall impression on the Discussion is that it merely repeats the results, and there is not a lot of elaboration and expansion. The results should be integrated and interpreted as a whole, there is no need to partialize finding by specific outcomes.

Line 539 – Fonagy’s gold standard – it is first time mentioned in the discussion….

Author Response

Response to Reviewer 3 Comments

Point 1: The manuscript presents the results of a valuable project, which deals with the problem of possible negative developmental outcomes in children at risk for maltreatment. Building on completed Phase 1 pilot data, Phase 2 examined several outcomes including parental reflective function and child development, as well as parental perceived social support and executive function, and children’s behaviour and executive function.

Although the manuscript is well written, in my opinion some changes could be made to make it clearer and easier to read.

The Introduction is too long. It needs to be shortened and more focused on specific goal of this research which is to determine the impact of ATTACHTM on parental RF and children’s development outcomes of interest.

Response 1: We have extensively edited the Introduction section as requested. We deleted excessive wordiness, as follows:

- Lines 50-53: “RF differs from related terms such as mindblindedness, mindreading, theory of mind, metacognition, mindfulness, empathy, and emotional intelligence [15, 17, 18]. While not synonymous with RF, they may tap the same underlying neurobiological socio-cognitive system as RF and focus on internal representations of the child [18-20].”

- Lines 58-64: “Parents who are highly reflective interpret their child’s behaviour by considering their own mental and those of their children as well as interactions between the each other’s mental states [10, 11, 13] while those with low RF tend to be unaware of their own internal experiences and/or their children’s mental states [13, 20]. Parental RF has been linked to parent-child interaction quality, with higher RF associated with sensitive parenting and lower RF associated with disruptive parenting quality [11, 26-30].”

- Lines 65-78: “The ability of parents to be reflective during interactions with their children is often observed to be suboptimal in populations at risk for maltreatment, including those affected by parental histories of early trauma or adversity, current drug and alcohol abuse, psychopathology, and/or dysfunctional parenting [3, 7, 31-33]. High-risk families are also likely to be exposed to intersecting socio-economic risks including low maternal education, single parenthood, young age at conception, and minority status [34]. Moderate and higher RF ability in mothers predicts secure infant attachment, in both low [10, 20, 35] and high-risk [32, 36] samples.

Parental RF appears modifiable by intervention [27, 28, 37-42], thus early intervention may foster secure attachment and healthy child development. A recent systemic review [37] of dyadic interventions targeting improvement in parental RF reported a significant reduction in disorganised attachment in infants (risk ratio: 0.50; 95% CI [0.27, 0.90]) [37]. Parental RF interventions also improved RF for mothers experiencing addiction [28, 38-41], imprisonment [43-45], depression, family violence, and/or poverty [11, 27, 46].”

- Lines 84-92: These mental abilities enable an individual to control their emotions and behaviour to achieve their goals [49, 50]. Compared to mothers with less optimal working memory, those with better working memory are more capable of managing their own emotions and behavioral responses to their distressed infants [51-55], display more interest in their child's feelings and curiosity [56], and more positive behaviours towards their children in a frustration-based task [57, 58]. Mothers with better set-shifting capacities show higher levels of interest and curiosity in their children's thoughts and feelings, and adjust their behaviors during dyadic interactions to accommodate their children, consistent with higher parental RF [59].”

Given the wide range of outcomes and theoretical complexities (see comment #2 below) to discuss, we could not identify further areas for deletion.

Point 2: Reflective function as a main construct of interest should be better explained and elaborated within some theory. A lot is written about RF and its relations to other constructs included in this study, but the theoretical frame is missing or should be more emphasized.

e.g. Lines 48 and 49 - RF differs from related terms such as mindblindedness, mindreading, theory of mind, metacognition, mindfulness, empathy and emotional intelligence. How? In what way? From the given definition, these constructs are quite similar.

Response 2: In light of reviewer’s 3 comment #1 above, we elected not to add text. However, if the editor, we can provide a table that could be included in an appendix describing the differences between the terms.

Point 3: I propose to the authors to shorten the introduction, in a way that they first give some theoretical frame for the examined constructs, short overview of the programs for interventions and their own program. Authors could shorten the description of research results for every outcome. There is a lot of text and information in the Introduction, which makes the text hard to follow.

Response 3: See responses to reviewer’s 3 comments 1 and 2 above. However, we added reference to Peter Fonagy (father of mentalization theory and RF) in our introduction on RF in lines 45-50 as follows. “Parental RF is defined as the parent’s (or primary caregivers’) ability to think about and identify thoughts, feelings, and mental states in themselves and in their children [13, 14]. According to leading theorist Peter Fonagy, RF operationalizes mentalizing which involves attending to mental states in oneself and others and interpreting behaviour accordingly [15, 16].”

Point 4: Participant and measures are adequately described, and statistical analyses are appropriate.

Response 4: Thank you for your great feedback.

Point 5: In the Results section, Tables 1-4 can be removed from the manuscript. Demographic information of the parents and children are described in the text and there is no need to show it again in the table. If authors think this information are vital, they can be presented at the end of the manuscript or in the supplement material.

Instead of these tables, it would be helpful to have one table with the important information about the design – the overview of the intervention with information on the participant at each phase and pilot studies, and outcomes of interest.

Response 5: In responding to reviewer’s 2, we kept the tables because they provide more information and deleted the redundant information from the text in lines 322-442.

Point 6: Please do not mark results with p = .053 as significant (lines 427 and Table 5).

Response 6: Thank you. We have made that change in Table 5.

Point 7: Discussion – please start this section with brief introduction, not with the results. The overall impression on the Discussion is that it merely repeats the results, and there is not a lot of elaboration and expansion. The results should be integrated and interpreted as a whole, there is no need to partialize finding by specific outcomes.

Response 7: Thank you. We revised it in lines 452-454 as follows. “This paper sought to examine the impacts of ATTACH™ on parental RF, perceived social support, and executive function as well as children’s development, behavioural and sleep problems, and executive function.”

Point 8: Line 539 – Fonagy’s gold standard – it is first time mentioned in the discussion….

Response 8: We alluded to Fonagy in the opening paragraph describing how RF operationalizes the concept of mentalizing – please see lines 45-50 as described above.

We also deleted the term gold standard as not essential in lines 475-476. We hope that these changes will make the link between Fonagy and the measurement of RF clearer, and no longer surprising when discussed in the discussion.

We hope that you find the changes to be satisfactory.
